# Molecular and Functional Verification of Wharton’s Jelly Mesenchymal Stem Cells (WJ-MSCs) Pluripotency

**DOI:** 10.3390/ijms20081807

**Published:** 2019-04-12

**Authors:** Aleksandra Musiał-Wysocka, Marta Kot, Maciej Sułkowski, Bogna Badyra, Marcin Majka

**Affiliations:** Department of Transplantation, Faculty of Clinical Immunology and Transplantation, Institute of Pediatrics, Collegium Medicum Jagiellonian University, Wielicka 265, 30-663 Kraków, Poland; aleksandra.musial@uj.edu.pl (A.M.-W.); marta.kot@uj.edu.pl (M.K.); maciek.sulkowski@uj.edu.pl (M.S.); bogna.badyra@doctoral.uj.edu.pl (B.B.)

**Keywords:** mesenchymal stem cells, pluripotency, OCT-4, Nanog, SSEA-4, iPS, WJ-MSC

## Abstract

The properties of mesenchymal stem cells (MSCs), especially their self-renewal and ability to differentiate into different cell lines, are widely discussed. Considering the fact that MSCs isolated from perinatal tissues reveal higher differentiation capacity than most adult MSCs, we examined mesenchymal stem cells isolated from Wharton’s jelly of umbilical cord (WJ-MSCs) in terms of pluripotency markers expression. Our studies showed that WJ-MSCs express some pluripotency markers—such as NANOG, OCT-4, and SSEA-4—but in comparison to iPS cells expression level is significantly lower. The level of expression can be raised under hypoxic conditions. Despite their high proliferation potential and ability to differentiate into different cells type, WJ-MSCs do not form tumors in vivo, the major caveat of iPS cells. Owing to their biological properties, high plasticity, proliferation capacity, and ease of isolation and culture, WJ-MSCs are turning out to be a promising tool of modern regenerative medicine.

## 1. Introduction

In general, the ‘stemness’ of cells can be defined as the most primitive cell state that combines two inseparable properties: the ability of self-renewal and the ability to differentiate. More accurately, stem cells (SCs) can generate daughter cells that are identical to their mother cells (self-renewal), as well as progenitor/precursor cells with more restricted potential (differentiation) [1,2,3,4].

The maintenance of stemness state requires presence of set of distinctive factors that are expressed in the cells. The transcription factors NANOG and OCT-4 are considered essential for maintaining the state of stem cell pluripotency [2,5,6,7,8,9,10]. NANOG (the name derives from Tìr na nÒg, the mythical Celtic land of youth) described by Chambers et al. 2003 and Mitsui et al. 2003 in mouse embryonic stem cells (ESCs) is recognized as the main factor responsible for stem cell self-renewal processes as well as their pluripotent nature [11,12,13]. Overexpression of this protein in embryonic cells increases their proliferative activity as well as maintains them in an undifferentiated state. OCT-4 (Octamer-binding transcription factor-3/4) is present primarily in early embryos where it plays a key role in the regulation of pluripotency and stem cell differentiation processes by regulation of other genes’ expression [1,2,9].

Mesenchymal stem cells (MSCs) are a type of multipotent stem cell that are located in many adult and fetal tissues. Apart from bone marrow, they have been identified and isolated from adipose tissue, synovium, skin, skeletal muscle, dental pulp, umbilical cord, placenta and other tissues, peripheral blood, cornea and retina, liver, and pancreas [14,15]. Some cell characteristics—such as the expression of specific markers, production of secretomes (cytokines, trophic factors etc.) or exosomes (mRNA, siRNA etc.), and potential to proliferate or differentiate—may differ depending on the MSCs source tissue [16,17,18,19,20]. That is why MSCs from various sources are being tested for availability, ease of separation, in vitro culture requirements and potential clinical applications [15,17,18,21,22,23]. Due to these criteria, umbilical cord mesenchymal stem cells have earned a great attention. Wharton’s jelly mesenchymal stem cells (WJ-MSCs) show the minimal criteria outlined by the International Society of Cellular Therapy: they adhere to plastic in standard culture conditions, display fibroblast-like morphology, have tri-lineage differentiation potential (they are able to differentiate into adipocytes, chondrocytes and osteoblasts), and express specific surface antigens. Populations of the WJ-MSCs express specific mesenchymal markers such as CD90, CD105, and CD73. The cells also reveal high level of adhesion receptors—integrin β1 (CD29) and hyaluronan receptor—CD44. Furthermore, WJ-MSCs are negative for hematopoietic and endothelial antigens: CD45, CD3, and CD34 [3,4,24].

WJ-MSCs might be advantageous for cell-based therapy because of their high self-renewal capacity in vitro, high plasticity, low immunogenicity, and immunomodulatory properties [25].

Induced pluripotent stem cells (iPS) are obtained by gene reprogramming techniques from adult, differentiated somatic cells (most often from fibroblasts). After induced expression of pluripotency transcription factors (*OCT-4*, *SOX2*, *KLF4*, and *C-MYC*), they become pluripotent. Their pluripotency is confirmed by expression of the pluripotency markers *OCT-4*, *NANOG*, *SOX2*, *SSEA1*, *SSEA3*, *SSEA4*, *TRA1-60*, *TRA1-81*, ability to form teratomas in vivo, and similarity with ESCs [21,26,27,28].

Fetal-tissue-derived MSCs exhibit some unique properties—such as ease of isolation and culture, immunomodulatory capacity, ability to self-regenerate and high proliferation rate, differentiation into several cell lineages, and lack of ethical problems—that make them the most promising tool in cell-based therapy. Since it is considered that MSCs derived from fetal tissues are more potent than from adult tissues, the main goal of the current study was to analyze the pluripotency of mesenchymal stem cells isolated from Wharton’s jelly. Additionally, we have compared potency of WJ-MSCs with reference to iPS as cells of well-defined pluripotent properties. It should be noted that the oxygen concentration is one of the key factors affecting metabolism, proliferation capacity, and phenotype of the cells [29,30,31,32,33,34]. Since hypoxia is the most similar to the physiological conditions occurring in living organisms, we decided to examine WJ-MSCs cultured both in normoxic and hypoxic conditions. This approach allows for better understanding biological properties of WJ-MSCs.

## 2. Results

### 2.1. Characterization of WJ-MSCs

Three different WJ-MSCs lines were cultured in vitro under normoxic and hypoxic conditions. No significant differences were observed at both conditions (Figure 1A–C).

Cytometric analysis confirmed the presence of WJ-MSCs specific markers characteristic for MSCs—namely CD73, CD90, and CD105—and a lack of expression for CD3 and CD45 antigens (Figure 1D,E). This data confirms the homogeneity of the cells’ populations. It should be noted that WJ-MSCs cultured under standard conditions and with low partial pressure of oxygen express the aforementioned markers at similar level. There are also no significant differences in expression level of investigated markers between different WJ-MSCs lines (Figure 1E).

Comparative analysis of transcriptome of investigated WJ-MSC’s population revealed a similarity of genes profile between three samples. However, the expression data showed that WJ-MSCs 2 and WJ-MSCs 3 are more similar compared to WJ-MSCs 1 (Figure 1F).

The genes depicted in Figure 1F were chosen based on the biggest differences and the highest similarities between the three WJ-MSCs cell lines characterized. Genes with the highest expression (Figure 1G, yellow), with medium expression (Figure 1G, orange) and with the lowest expression (Figure 1G, red) in all three lines of WJ-MSCs are shown. Genes with different expression between tested WJ-MSCs lines are marked in blue color.

### 2.2. Pluripotency Marker Expression in Normoxic and Hypoxic Conditions

The analysis of *OCT-4* and *NANOG* expression at the mRNA level by real-time qPCR demonstrated lower level of the mRNA expression in WJ-MSCs under both low and high oxygen concentrations compared to iPS cells (Figure 2A,B). However, our data confirm previous results that, at the mRNA level, the expression of *NANOG* under hypoxic conditions is increased (Figure 2A).

Intracellular analysis of NANOG and OCT-4 using flow cytometry revealed low expression or no expression of both proteins in WJ-MSCs (Figure 2C,D). However, hypoxic conditions stimulated NANOG expression in larger populations of cells (Figure 2D). The presence of OCT-4 did not change or increase only slightly. As expected, more than 90% of the iPS cells express both factors (Figure 2D). The genetic modification in case of WJ-MSCs Luc+ has no influences on NANOG and OCT-4 expression (Figure 3D,E).

The three WJ-MSCs preparations were analyzed using Lyoplate screening panels. The WJ-MSCs did not express stage-specific embryonic antigens 3 (SSEA-3) and tumor rejection antigen (TRA 1-60) markers but they expressed SSEA-4 at relatively high level in all 3 lines of WJ-MSCs (Figure 2E). The level of SSEA-4 expression was slightly decreased by culture of WJ-MSCs in 5% of oxygen concentration. The expression of SSEA-3 and TRA 1-60 was not influence by low oxygen concentration (Figure 2E). At the same time iPS cells showed strong presence of SSEA-3, SSEA-4, and TRA 1-60 (Figure 2C,E). The expression level of surface markers has been estimated in reference to appropriate isotype control.

The flow cytometry results were confirmed by additional immunocytochemical staining (Figure 3). As expected, iPS cells showed high expression of all the pluripotency markers (Figure 3A). SSEA-4 was detected in WJ-MSCs cultured at both 5% and 21% oxygen concentrations (Figure 3B). However, when WJ-MSCs were cultured in low oxygen concentrations, expression of NANOG and OCT-4 was observed (Figure 3C).

### 2.3. In Vivo Safety of WJ-MSCs—Tumorogenicity Assay

In order to check the tumorigenicity of WJ-MSCs, they were marked with Luciferase, cultured in normoxia and hypoxia and injected subcutaneously into immunodeficient NOD-SCID mice. Mice were observed for as long as 3 months and imaged with the Optical Preclinical Imaging System. During the experiment, gradual loss of the luminescence signal was observed in both groups. After 30 days, the luminescence signal was lost in the normoxia group (Figure 4A–C) but in hypoxia group it was still observable until day 37 (Figure 4E–G). After 90 days luminescence signal from both injected groups was not detected (Figure 4D,H). During the experiment, no deterioration in the health of mice was observed, as well as noticeable changes in the site and area of the implanted WJ-MSCs. Thus, after transplantation of WJ-MSCs, there was no tumorigenic proliferation observed. At the same time, in group consisted of mice injected with iPS Luc^+^ cells the signal of luminescence slightly decreased after a few days (Figure 4J) but on day 24 after administration, an increase of the luminescence signal was observed (Figure 4K) which persisted until day 30 when mice were killed (Figure 4L). Tumors from mice in the iPS Luc^+^ group were isolated and fixed (Figure 4M). Luminescence imaging analysis of the tumors confirmed their derivation from Luc positive cells (Figure 4N) and histopathological analysis revealed very characteristic morphology of teratomas that included tissues from all germ layers (Figure 4O).

## 3. Discussion

The specific properties of the mesenchymal stem cells, particularly their ability to differentiate and their paracrine activity, are of the special interest in terms of their therapeutic effects. In preclinical and clinical studies, their beneficial effect has been demonstrated in the treatment of many diseases: cardiovascular diseases [35], neurodegenerative diseases [36], immune-mediated diseases [37], etc. The MSCs are commonly classified as multipotent, but they express a relatively low level of pluripotent markers characteristic for embryonic stem cells (ESCs) and iPS cells. It has been demonstrated that pluripotent cells express a unique set of factors necessary for the maintenance of stemness—namely OCT-4, NANOG, and SOX2 [2,5,6,7,8,9]. As described previously, these transcription factors are involved in regulation of the pluripotency, self-renewal, and proliferation [7]. *OCT-4* is necessary to initiate embryonic development of mammals and is essential for the formation of embryos’ inner cell mass and ESCs maintenance [8]. *SOX2* regulates expression of *OCT-4* and maintains the pluripotent state of ESCs, whereas *NANOG* is required for the maintenance of undifferentiated state and self-renewal of stem cells [7]. The same factors are also responsible for the pluripotent state of iPS cells.

Our studies have shown that WJ-MSCs express pluripotency markers: NANOG, OCT-4, and SSEA-4. Our results are consistent with those reported by other investigators who also described the presence of OCT-4, SOX2, and NANOG in mesenchymal stem cells isolated from human umbilical cord [10,24,38]. However, our data show that WJ-MSCs exhibit much lower expression levels of these markers in comparison to human iPS cells. Similar results were obtained for ESCs and WJ-MSCs by Gao et al. They demonstrated significant expression of NANOG, SSEA-4, and SOX2, but the level of expression was relatively lower in WJ-MSCs than in human ESCs [10].

The analysis of transcriptome profiles of all examined cells confirmed that investigated WJ-MSCs are highly homogeneous population of cells. In the present study we selected for analysis group of genes with the highest and the lowest expression. Our results revealed that in group of genes with the highest expression, there are no genes known as pluripotency markers (*OCT-4*, *NANOG*, *SOX2*, *SSEA-1*, *SSEA-3*, *SSEA-4*, *TRA1-60*, *TRA1-81*). These results may explain why WJ-MSCs do not induce teratomas when transplanted into immunodeficient mice.

The culture conditions are very important issue that should be considered during examination of properties of the cells. Microenvironment, including oxygen conditions, is one of the main factors affecting metabolism, angiogenesis, proliferation, differentiation, and stemness of cells [29,30,31,32,33,34,39]. In this study we have observed strong influence of hypoxic conditions on expression of pluripotency markers, especially NANOG, at both mRNA and protein level. Our results are in partial agreement with data presented by Drela and colleagues, who showed that WJ-MSCs cultured in 21% O_2_ do not express *OCT-4*, *NANOG*, *SOX2*, and *REX 1* by immunocytochemistry staining and RT-PCR analysis. However, they also showed that 5% oxygen promotes expression of these genes [31]. This discrepancy may be due to WJ-MSCs isolation protocol and/or culture conditions. Apart from its effect on expression of pluripotency factors, hypoxia also affects proliferation rate of WJ-MSCs. Nekanti and colleagues [39] showed that WJ-MSCs under hypoxic conditions have higher proliferation rate and shorter mean population doubling time compared to WJ-MSCs cultured in normoxia. Our observations did not confirm that, since we observed only slightly higher proliferation rate of WJ-MSCs in hypoxia than in normoxia, which can be explained by the expression of NANOG already in normoxic conditions.

ESCs and iPS, that express the pluripotency genes form teratomas what proves their pluripotent character [40]. Considering the fact that WJ-MSCs express some of the pluripotency genes, we also examined them in terms of tumorigenicity. Since the expression of OCT-4 and NANOG increased in low oxygen conditions we transplanted WJ-MSCs cultured in normoxic or hypoxic environments. Interestingly, we did not observe any tumor development even after 90 days after transplantation, despite the fact that these cells display intermediate potency between pluripotent stem cells and adult MSCs [17,41]. iPS cells injected at the same time generated teratomas in vivo three weeks after injection. These results clearly show that WJ-MSCs are not tumorigenic despite expression of pluripotent genes. Thus, the cell-based therapy involving WJ-MSCs seems to be safe from that standpoint.

It has been proved that stem cells may migrate and engraft in the site of injury. Their repair mechanism is demonstrated by the secretion of many factors including cytokines and extracellular vesicles that affect tissue regeneration. Therefore, many studies focus on cell survival after transplantation. Toma et al. (2002) demonstrated the presence of transplanted MSCs in the mice hearts for up to 4 days [42]. In the porcine model of myocardial ischemia, it was observed that the injected cells are able to survive 10 days after administration [43]. McGinley et al., 2013 have reported that MSCs were detectable in an infracted rat heart at a very low level 28 days after transplantation [44]. Our studies have shown that transplanted WJ-MSCs cultured under hypoxia conditions are detectable at 37 days after injection. The survival time of cells that were normally grown under normoxia conditions is clearly shorter what does not exclude their regenerative properties. It is commonly known that stem cells derived from adult tissues exhibit much lower potency than embryonic stem cells [45]. Interestingly, WJ-MSCs derived from fetal tissue (human umbilical cord) not only possess mesenchymal stem cells properties, but also some features characteristic for ESCs [22,23,46,47,48]. Since it is hypothesized that WJ-MSCs are formed within the mucous connective tissue of the umbilical cord (Wharton’s jelly) in the early days of embryonic development and then reside there during gestation [49], this may explain their embryonic character and greater potential than MSCs’ derived from adult sources.

It is believed that MSCs are pluripotent/multipotent, thus able to differentiate into three germ layer cell types [6,50,51]. They easily differentiate into adipocytes, chondrocytes and osteocytes, but there are also reports about differentiation of MSCs into other germ layer derivatives such as neurons (derived from ectoderm) [51] and hepatocytes (originated in endoderm). However, non-mesodermal differentiation of MSCs is still discussed because of insufficient in vivo evidence [52]. Since we were unable to growth teratomas from WJ-MSCs, our data exclude the pluripotency-based phenotypic differentiation characterization of WJ-MSCs despite expression of OCT-4, NANOG and SOX2.

## 4. Materials and Methods

### 4.1. Collection, Isolation, and Expansion of WJ-MSCs

The umbilical cords were collected from three patients after Caesarean sections after parents’ consent. After washing with phosphate-buffered saline (PBS) solution supplemented with antibiotic-antimycotic solution, umbilical cords were cut into small (5 mm) pieces. Around 50 small slices of umbilical cord were put onto the plastic flask. Explants were cultured with growth medium for mesenchymal stem cells (DMEM Low Glucose, Biowest, USA), supplemented with platelet lysate in a standard culture conditions under 21% of O_2_ and 5% of CO_2_ at 37 °C and hypoxic condition under 5% of O_2_ and 5% of CO_2_ at 37 °C. After 7 days, the explants were removed, and the cells were passaged using Accutase cell detachment solution (BioLegend, USA). After reaching the appropriate number of cells, WJ-MSCs were cultured in the Quantum Cell Expansion Bioreactor (Therumo BCT), then frozen and thawed for analysis at passage 5. Isolated colonies of cells displayed fibroblast-like morphology. Inverted light microscope Olympus IX70 with phase contrast optics equipped with a Canon EOS1100D digital photo camera was used for routine observation, analysis, and documentation of the MSCs at a morphological level.

The three cell lines (WJ-MSCs 1, WJ-MSCs 2, WJ-MSCs 3) examined in this study have been isolated from three umbilical cords (from three different donors).

The cells were obtained as part of the project STRATEGMED II (agreement no. Strategmed2/265761/10/NCBR/2015).

### 4.2. iPS Cell Cultures

Protein induced pluripotent stem cells (piPS, System Biosciences, USA), were cultured on Gelatin (Sigma-Aldrich, St. Louis, MO, USA) coated dishes with feeder layer of mitomycin C (Sigma-Aldrich, USA) inactivated mouse embryonic fibroblasts (MEFs) in serum-free iPS medium DMEM/F12 supplemented with 20% KSR, 2 mM GLUTAMAX, 100 µM non-essential amino acids, 100 U/mL/100 µg/mL Penicillin/Streptomycin, 10 ng/mL bFGF (all from ThermoFisher Scientific, USA) and 100 µM -mercaptoethanol (Sigma-Aldrich). The medium was changed every day. When the culture reached about 80% confluence, iPS cells were subcultured with Accutase (Lonza, Basel, Switzerland) in proportion 1:4–1:10 and seeded onto fresh feeder layer in presence of 10 µM ROCK inhibitor Y-27632 (Sigma-Aldrich). For immunocytochemical stainings, iPS cells were cultured in feeder-free conditions on growth factor-reduced Matrigel (Corning, New York, NY, USA) coated dishes in MEF-conditioned iPS medium. MEFs were cultured in Dulbecco’s modified Eagle’s medium (DMEM; 4.5 g/L glucose; Lonza), supplemented with 10% *v*/*v* FBS, 2 mM L-glutamine and 100 U/mL/100 µg/mL Penicillin/Streptomycin antibiotics solution (all from ThermoFisher Scientific, Waltham, MA, USA).

### 4.3. Flow Cytometry Analysis

#### 4.3.1. Immunophenotype Analysis

Expression of specific surface antigens outlined for mesenchymal stem cells by the International Society of Cellular Therapy on WJ-MSCs was tested using flow cytometry technique. The cells were harvested with Accutase cell detachment solution (BioLegend, San Diego, CA, USA). The WJ-MSCs were incubated with fluorochrome-conjugated (fluorescein isothiocyanate-FITC and phycoerythrin-PE) antibodies against CD73, CD90, CD105, CD3, and CD45 (Becton Dickinson, Franklin Lakes, NJ, USA) for 30 min. at 4 °C in darkness. Corresponding isotype antibodies were used as the control to exclude non-specific binding. The results were analyzed using Attune NxT Software v2.2 on Attune Nxt Flow cytometer (Thermo Fisher Scientific).

#### 4.3.2. Lyoplate Screening Panel

The expression level of adhesion molecules was evaluated using Lyoplate technology from BD (Becton Dickinson). The WJ-MSCs were stained with monoclonal primary antibodies and, subsequently with secondary antibodies conjugated with fluorophore against selected surface antigens from a Lyoplate screening panel. The procedure was carried out according to manufacturer’s instruction. The cells were acquired using Attune Nxt Flow Cytometer equipped with autosampler, that allows fully automated sample acquisition from 96-well plates. A total of 10,000 viable cells (events) were analyzed per sample. The data were analyzed and presented as standard histograms.

#### 4.3.3. Expression of Pluripotent Markers

WJ-MSCs were stained with fluorescein-isothiocyanate (FITC) and phycoerythrin (PE) conjugated primary antibodies against pluripotency markers SSEA-3, SSEA-4, TRA 1-60 (Becton Dickinson) for 30 min at 4 °C. Appropriate PE-conjugated or FITC-conjugated antibodies were used as an isotype control. The fluorescence intensity of the cells was evaluated by flow cytometry Attune Nxt Flow Cytometer. Data was analyzed using Attune NxT Software v2.2.

#### 4.3.4. Intracellular Staining

Harvested cells were washed with PBS and fixed with 4% paraformaldehyde (PFA, Sigma-Aldrich) for 15 min. Next, the cells were washed and permeabilized with 0.1% Triton X-100 (Sigma-Aldrich) for 20 min at room temperature. The cells were stained with primary antibodies anti-OCT-4 (1:100, Merck, Darmstadt, Germany) and anti-NANOG (1:500, Merck) and incubated for 1 h at RT. Then, the pellet of cells was stained with appropriate secondary antibody conjugated with Alexa Fluor-488 (1:200, Life Technologies, Carlsbad, CA, USA) for 60 min. The results were collected by flow cytometer Attune Nxt Cytometer and data was analyzed using Attune NxT Software v2.2.

### 4.4. Immunocytochemistry

The immunocytofluorescence staining method was used to determine the expression of the markers of particular cell types at the protein level. The WJ-MSCs and modified WJ-MSCs Luc^+^ were plated on 24-well plates on glass coverslips. When the cells were ready for staining, the culture medium was drained and the cells were fixed with 4% paraformaldehyde for 20 min at room temperature. The next step was permeabilization of cell membranes with 0.1% Triton X-100 solution in PBS (5 min at room temperature). After three washes with PBS, the sites of non-specific antibody binding were blocked using a 3% BSA solution (in PBS) for 30 min at room temperature. The cells were then incubated overnight at 4 °C with the appropriate primary antibody diluted in 3% BSA. The following day, the cells were washed three times with PBS and incubated at room temperature in the dark for 1 h with the appropriate secondary antibody diluted in 3% BSA and with Hoechst 33342 (nucleic acid binding) dye. The unbound antibody was then washed three times with PBS. Stained cells were compared to isotype control to confirm the specificity of staining and to distinguish staining from autofluorescence of cells.

### 4.5. Real-Time PCR (qPCR), RNA Isolation, Reverse Transcription

Total RNA was isolated using RNA GeneMATRIX Universal RNA Purification Kit (EURx, Gdańsk, Poland) according to the manufacturer’s instruction. Reverse transcription was performed using commercial reagent kit MMLV Reverse Transcriptase (Promega, Madison, WI, USA). The expression of the pluripotency factors *OCT-4* and *NANOG* at the mRNA level was verified by qPCR using SybrGreen (EURx) and designed primers (Table 1) and blank qPCR Master Mix reagent (EURx, Poland). The mRNA expression level for all samples were normalized to the housekeeping gene GAPDH transcript as a reference gene. Levels of mRNA expression were determined with the 2^−ΔΔ*C*t^ method. The real-time qPCR was performed with the Applied Biosystems QuantStudio 7 Flex real-time PCR system.

### 4.6. RNA Preparation for Whole-Genome Sequencing

Total RNA was extracted from WJ-MSCs samples using QIAGEN RNeasy Mini-Kit (QIAGEN, Hilden, Germany). RNA from all samples was run on an Agilent TapeStation System (Agilent Technologies, Santa Clara, CA, USA) to assess quality, and to estimate RNA concentrations, RNA was quantitated by Promega QuantiFluor Dye System on Quantus Fluorometer (Promega).

### 4.7. Gene Expression Quantification

300 ng of total RNA was used to generate biotinylated cRNA using the TargetAmp-Nano Labeling Kit for Illumina Expression BeadChip (Epicentre-an Illumina Company, Madison, WI, USA) which was fragmented and hybridized to an Illumina Whole Genome Expression Chip, HumanHT-12 v3.0. BeadChips were then washed and stained and subsequently scanned to obtain fluorescence intensities. Expression profiles were generated by hybridizing 750 ng of cRNA to Illumina HumanHT-12 v3.0 BeadChips according to Illumina whole-genome gene expression direct hybridization assay guide (Illumina).

### 4.8. Tumorogencity Assay (In Vivo)

Determination of the safety of the use of WJ-MSCs based therapy was performed with immunocompromised NOD-SCID mice. The experiment was conducted in two independent groups consisting of four mice per group. The groups got WJ-MSCs cultured in hypoxia and normoxia conditions respectively. Each mouse of two groups was injected subcutaneously with 2x10^6^ cells transduced with the vector harboring the luciferase gene (Luc^+^). As a positive control, mice were injected with 2x10^6^ iPS Luc^+^. The luminescence signal was measured biweekly in both groups until the signal was lost (optical preclinical imaging system/for small animals—Ami). Tumor formation in mice was monitored for 3 months. After this time, the luminescence signal was measured in the WJ-MSCs groups. Mice from WJ-MSCs groups were sacrificed after 3 months. In the control group after the appearance of tumors, the mice were sacrificed and the tumors were excised. Fixed tissues were imbedded in paraffin, cut, stained with hematoxylin/eosin and analyzed histopathologically. Each of the experiments was repeated three times in order to authenticate the obtained results.

The experiments on animals have been carried out in accordance with the “Guidelines for the Care and Use of Laboratory Animals”. The protocol was approved by the Second Local Institutional Animal Care and Use Committee (IACUC) in Krakow by decision number 278/2015 on 15 December 2015 and 162/2015 on 24 June 2015.

### 4.9. Statistical Analysis

Statistical analysis was performed with GraphPad Prism 7 software. Data are shown as ± standard error of the mean. Statistical comparisons were evaluated using an analysis of ANOVA test with post-hoc Dunnett’s test, with *p* ≤ 0.0001 for NANOG and *p* ≤ 0.0006 for OCT-4 considered statistically significant difference.

## 5. Conclusions

WJ-MSCs show all properties characteristic for mesenchymal stem cells outlined by the International Society for Cellular Therapy. Moreover, they also express some pluripotency markers, but compared to iPS cells, the expression of NANOG, OCT-4, and SSEA-4 is lower and SSEA-3 and TRA 1-60 do not occur. The better understanding of plasticity/potential of MSCs is very important in the context of their use in therapeutic purposes. The advantages of WJ-MSCs in clinical application include ease and high yield of isolation, proliferation, and immunomodulation. Furthermore, they are exempt from ethical implications of ESCs and their acquisition does not pose a risk of complications for the donor [4,53,54]. Importantly, despite their high proliferation potential and pluripotent gene expression, they do not form teratomas in vivo also after culture in hypoxic conditions. Thus, our data provide further evidence on the safe use of WJ-MSCs in clinical settings.

## Figures and Tables

**Figure 1 ijms-20-01807-f001:**
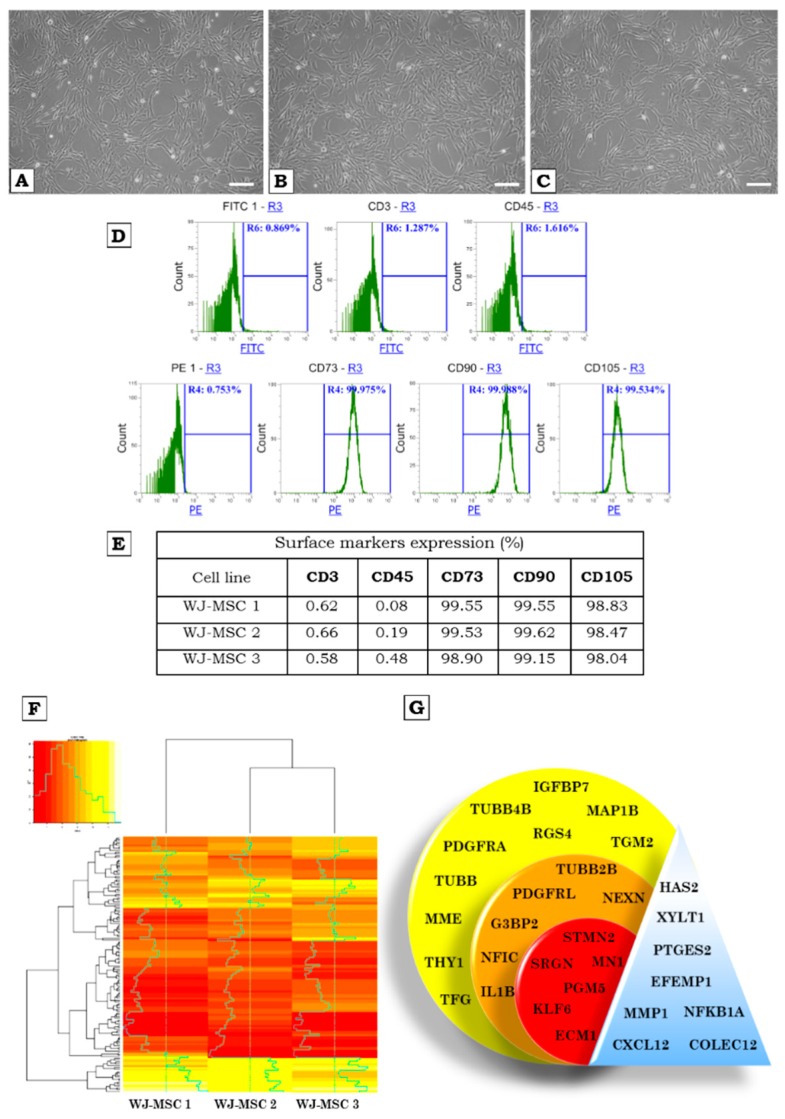
(**A**–**C**)—The cultures of WJ-MSCs, fifth passage. The cells grow in a monolayer, show adherent properties and fibroblast-like morphology; Morphology of cells cultured in (**A**–**C**)—normoxia—magnification 100×, white bars correspond to 50 µm. (**D**)—Flow cytometry analysis of surface markers expression of WJ-MSCs. The WJ-MSCs were labelled with anti-CD3-FITC and anti-CD45-FITC antibodies (negative markers) and anti-CD73-PE, anti-CD90-PE, anti-CD105-PE antibodies (positive markers), and analyzed by flow cytometry. (**D**)—Representative histograms showing expression of set of markers for WJ-MSCs (detailed analysis presented in the table—(**E**). (**F**)—a heatmap showing the transcription profile of investigated WJ-MSCs. (**G**)—transcripts of selected genes expressed in the WJ-MSCs. The colors of the circles correspond to the colors on the heatmap (**F**): yellow—high expression; orange—medium expression; red—low expression; triangle—genes with different expression level.

**Figure 2 ijms-20-01807-f002:**
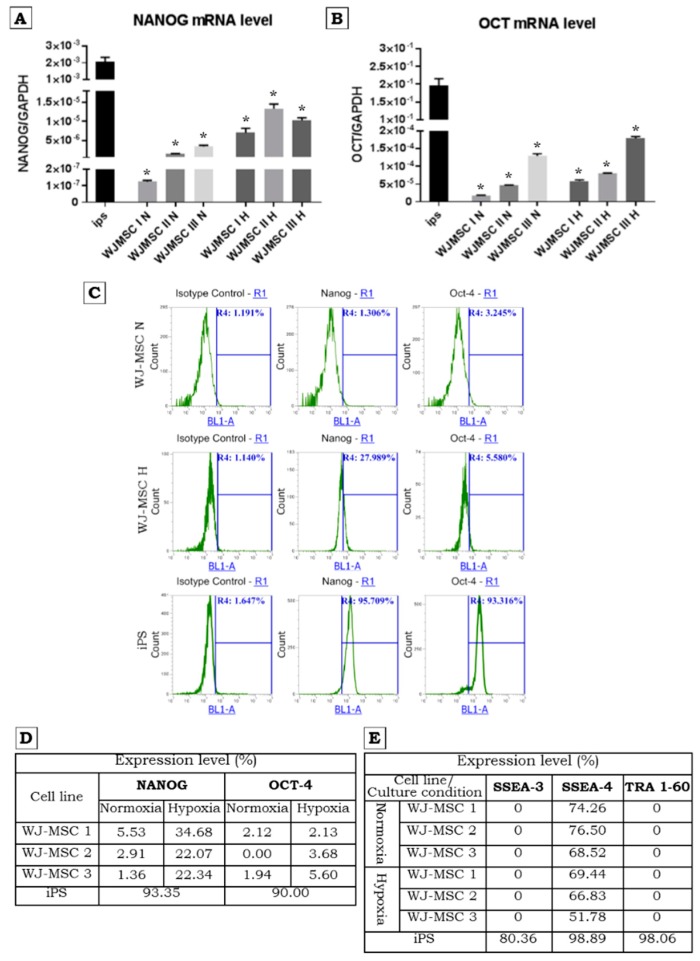
Pluripotency markers expression on mRNA (**A**,**B**) and protein level (**C**,**D**) determined with RT-qPCR and flow cytometry (intracellular staining) respectively. (**A**,**B**)—Comparison of NANOG and OCT-4 relative expression level by RT-qPCR in WJ-MSCs and iPS. iPS exhibits relatively very high expression compared to WJ-MSCs. Note that hypoxia promotes NANOG expression in WJ-MSCs. Results are presented as means ± standard error, * *p* ≤ 0.0001 (**A**); * *p* ≤ 0.0006 (B). (**C**,**D**)—expression of NANOG and OCT-4 examined using intracellular staining and flow cytometry analysis. The iPS show a high expression of NANOG and OCT-4. In the case of WJ-MSCs the visible higher expression of NANOG was noticed in hypoxic conditions. H—hypoxia, N—normoxia. (**E**)—Expression level of pluripotency markers: SSEA-3, SSEA-4, TRA 1-60 (antibody labelling and flow cytometric analysis). Note that iPS exhibit high expression all of examined pluripotency markers whereas on WJ-MSCs only SSEA-4 is expressed. H—hypoxia; N—normoxia.

**Figure 3 ijms-20-01807-f003:**
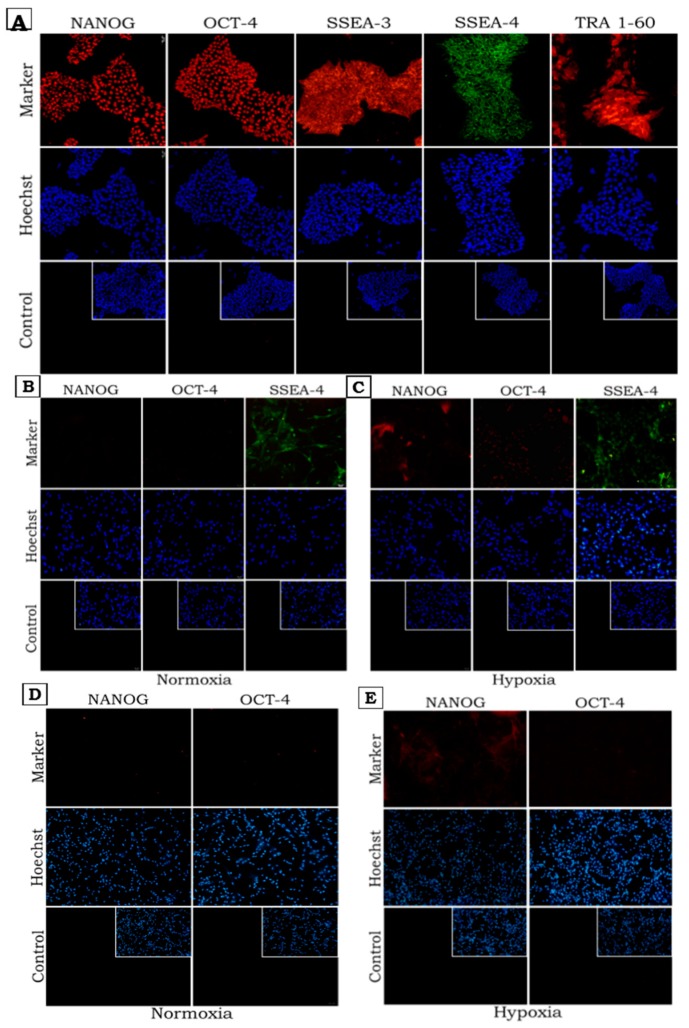
Immunofluorescence staining demonstrate distribution of pluripotency markers: NANOG, OCT-4 (transcription factors) and SSEA-3, SSEA-4, TRA 1-60 (surface markers) in iPS (**A**) and WJ-MSCs (**B**,**C**). (**D**,**E**)—expression of NANOG and OCT-4 in WJ-MSC Luc+ (the modified cell line expressing luciferase gene). Nuclear staining–Hoechst 33342: blue; NANOG and OCT-4 –Alexa Fluor 555: red; SSEA-4-FITC: green; TRA 1-60-PE, SSEA-3-PE: red. (**A**–**E**)—Florescence microscope, (**A**) magnification ×150, insets- magnification ×100; (**B**–**E**)—magnification ×100, insets—magnification ×80.

**Figure 4 ijms-20-01807-f004:**
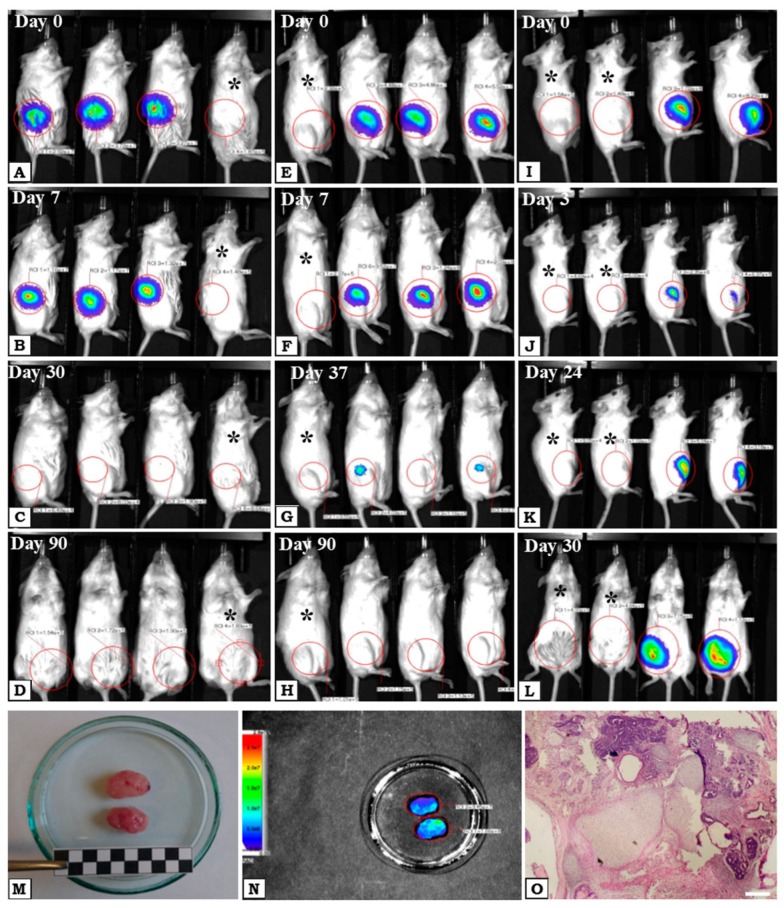
NOD-SCID mice injected subcutaneously with cells transduced with the vector transferring the luciferase gene (Luc): WJ-MSCs Luc+ (**A**–**H**) and iPS Luc+ (**I**–**L**). The modified WJ-MSCs were cultured in two oxygen conditions: WJ-MSCs Luc+ normoxia (**A**–**D**) and WJ-MSCs Luc+ hypoxia (**E**–**H**). The presence of injected cells was observed after luciferin administration by measuring the bioluminescence signal. After 30 days, in mice with normoxia, WJ-MSCs Luc+ signal was not detectable (**C**), whereas in mice with WJ-MSCs Luc+ hypoxia signal persisted. In mice with iPS Luc+ luminescence signal increased as a result of tumor development (**G**,**H**). 30 days after transplantation the tumors were isolated (**M**,**N**) and subsequently analyzed with hematoxylin and eosin staining (**O**). (**O**)—Light microscopy, magnification ×100, white bar corresponds to 150µm. Black asterisks—control mice without injected cells.

**Table 1 ijms-20-01807-t001:** Primer sequences used for real-time PCR genes expression analysis.

Gene	Forward Primer	Reverse Primer
*OCT-4*	ATGGCGGGACACCTGGCTT	GGGAGAGCCCAGAGTGGTGACG
*NANOG*	TGAACCTCAGCTACAAACAG	TGGTGGTAGGAAGAGTAAAG
*GAPDH*	CAAAGTTGTCATGGATGACC	CCATGGAGAAGGCTGGGG

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
