# Peer review of "Molecular and Functional Verification of Wharton’s Jelly Mesenchymal Stem Cells (WJ-MSCs) Pluripotency"

_ijms, 2019, doi:10.3390/ijms20081807_

Reviewer 1 Report

The study led by a well-established PI was insightfully designed and efficiently executed. The results were clearly presented and paper effectively written. The findings carry excellent values for both basic and translational investigations. There are only some minor points that can be readily addressed. Overall, this is an excellent paper for the journal's readership spectrum.

Minor issues:

1. Since WJ-MSCs express significantly lower levels of some pluripotency markers comparing to iPS cells, consider to modify the title from “Molecular and Functional Validation of Wharton’s Jelly Mesenchymal Stem Cells (WJ-MSCs) pluripotency” to “Molecular and Functional Verification of Wharton’s Jelly Mesenchymal Stem Cells (WJ-MSCs) pluripotency”

2. Consider adding “SSEA-4” as a keyword, given its unique feature in both PSCs and MSCs, and detection in WJ-MSCs

3. Since Oct4 was spelled out consider pointing out that the nomenclature of NANOG, a homeobox gene initially identified in mESCs (Chambers et al. 2003 and Mitsui et al. 2003), was derived from Tìr nan Òg (land of youth), one of the names for the Celtic Otherworld (Smyth, Daragh. A guide to Irish mythology (2nd ed.) 1996. Dublin: Irish Academic Press. ISBN 0716526123. OCLC 36338076)

4. For Lines 54-57, consider modifying to – “such as the expression of specific markers, production of secretomes (cytokines, trophic factors etc.) or exosomes (mRNA, siRNA, etc.), and potential to proliferate or differentiate may differ…” (Suggested references: Teng YD, et al. 2011. Functional multipotency of stem cells: a conceptual review of neurotrophic factor-based evidence and its role in translational research. Curr Neuropharmacol. 9(4):574-85. Teng YD. 2019. Functional multipotency of stem cells: Biological traits gleaned from neural progeny studies. Semin Cell Dev Biol. pii: S1084-9521(18)30059-4)

5. In Lines 59-60, reassess correctness of the following statement – “…(WJ-MSCs) show the minimal criteria outlined by…”

6. Lines 71-73: Consider changing “Their pluripotency is confirmed by expression of the pluripotency markers OCT-4, NANOG, SOX2, SSEA1, SSEA3, SSEA4, TRA1-60, TRA1-81 and ability to form teratomas in vivo [11-13].” to “Their pluripotency is confirmed by expression of the pluripotency markers OCT-4, NANOG, SOX2, SSEA1, SSEA3, SSEA4, TRA1-60, TRA1-81, ability to form teratomas in vivo, and similarity with ESCs [refs].” (one suggested reference: Parsons XH, et al. 2011. Efficient derivation of human cardiac precursors and cardiomyocytes from pluripotent human embryonic stem cells with small molecule induction. J Vis Exp. 57:e3274)

7. Lines 74-75: consider changing “it is considered that WJMSCs derived from fetal tissues are more potent than adult stem cells,” to “…than adult tissues…”

8. Lines 184-185:  Change “In mice with iPSs Luc+ luminescence signal increased what was a result of tumor development” to “In mice with iPSs Luc+ luminescence signal increased as a result of tumor development”

9. For Fig. 4, consider only including mice with donor cell mass detectible in Day 0, and verbally explain inclusion criteria (current Fig. 4 with mice lacking donor cell mass images in Day 0 may cause confusion)

10. Line 202: Consider changing “Our results are conclusive with other authors’ who…” to “Our results are consistent with those reported by other investigators that…”

11. Line 250: Consider changing “our data exclude the pluripotent character of WJMSCs despite expression of OCT-4, NANOG and SOX2” to “our data exclude the pluripotency-based phenotypic differentiation characterization of WJMSCs despite expression of OCT-4, NANOG and SOX2.”

Author Response

1. Since WJ-MSCs express significantly lower levels of some pluripotency markers comparing to iPS cells, consider to modify the title from “Molecular and Functional Validation of Wharton’s Jelly Mesenchymal Stem Cells (WJ-MSCs) pluripotency” to “Molecular and Functional Verification of Wharton’s Jelly Mesenchymal Stem Cells (WJ-MSCs) pluripotency”

This title of the publication has been changed according to the recommendation of the reviewer.

2. Consider adding “SSEA-4” as a keyword, given its unique feature in both PSCs and MSCs, and detection in WJ-MSCs

SSEA-4 as a keyword was added.

3. Since Oct4 was spelled out consider pointing out that the nomenclature of NANOG, a homeobox gene initially identified in mESCs (Chambers et al. 2003 and Mitsui et al. 2003), was derived from Tìr nan Òg (land of youth), one of the names for the Celtic Otherworld (Smyth, Daragh. A guide to Irish mythology (2nd ed.) 1996. Dublin: Irish Academic Press. ISBN 0716526123. OCLC 36338076)

This part of the publication has been changed according to the recommendation of the reviewer.

4. For Lines 54-57, consider modifying to – “such as the expression of specific markers, production of secretomes (cytokines, trophic factors etc.) or exosomes (mRNA, siRNA, etc.), and potential to proliferate or differentiate may differ…” (Suggested references: Teng YD, et al. 2011. Functional multipotency of stem cells: a conceptual review of neurotrophic factor-based evidence and its role in translational research. Curr Neuropharmacol. 9(4):574-85. Teng YD. 2019. Functional multipotency of stem cells: Biological traits gleaned from neural progeny studies. Semin Cell Dev Biol. pii: S1084-9521(18)30059-4)

This part of the publication has been changed according to the recommendation of the reviewer.

5. In Lines 59-60, reassess correctness of the following statement – “…(WJ-MSCs) show the minimal criteria outlined by…”

This part of the publication has been corrected according to the recommendation of the reviewer.

6. Lines 71-73: Consider changing “Their pluripotency is confirmed by expression of the pluripotency markers OCT-4, NANOG, SOX2, SSEA1, SSEA3, SSEA4, TRA1-60, TRA1-81 and ability to form teratomas in vivo [11-13].” to “Their pluripotency is confirmed by expression of the pluripotency markers OCT-4, NANOG, SOX2, SSEA1, SSEA3, SSEA4, TRA1-60, TRA1-81, ability to form teratomas in vivo, and similarity with ESCs [refs].” (one suggested reference: Parsons XH, et al. 2011. Efficient derivation of human cardiac precursors and cardiomyocytes from pluripotent human embryonic stem cells with small molecule induction. J Vis Exp. 57:e3274)

This part of the publication has been changed according to the recommendation of the reviewer.

7. Lines 74-75: consider changing “it is considered that WJMSCs derived from fetal tissues are more potent than adult stem cells,” to “…than adult tissues…”

This part of the publication has been changed according to the recommendation of the reviewer.

8. Lines 184-185:  Change “In mice with iPSs Luc+ luminescence signal increased what was a result of tumor development” to “In mice with iPSs Luc+ luminescence signal increased as a result of tumor development”

This part of the publication has been changed according to the recommendation of the reviewer.

9. For Fig. 4, consider only including mice with donor cell mass detectible in Day 0, and verbally explain inclusion criteria (current Fig. 4 with mice lacking donor cell mass images in Day 0 may cause confusion)

The mice lacking donor cell mass are the control group without administration the cells. They are marked by black asterisks. Mice without the injected cells (control mice) were observed during experiment simultaneously with injected mice to  cut off the signal background.

10. Line 202: Consider changing “Our results are conclusive with other authors’ who…” to “Our results are consistent with those reported by other investigators that…”

This part of the publication has been changed according to the recommendation of the reviewer.

11. Line 250: Consider changing “our data exclude the pluripotent character of WJMSCs despite expression of OCT-4, NANOG and SOX2” to “our data exclude the pluripotency-based phenotypic differentiation characterization of WJMSCs despite expression of OCT-4, NANOG and SOX2.”

This part of the publication has been changed according to the recommendation of the reviewer.

Reviewer 2 Report

The article addresses an interesting issue of the importance of pluripontency markers for the function of Wharton’s Jelly-derived mesenchymal stem cells that in fact by definition are not pluripotent. Due to the advantages of this cell source (ease of access, high potency) there is ample interest in utilizing them clinically. Overall this paper is informative and conclusions are in most cases supported by presented data several deficiencies should be corrected to improve clarity and scientific value of this work. Specific comments are listed below:

At the end of introduction authors indicate WJMSCs as more potent than adult stem cells. Please specify whether it is differentiation repertoire, proliferative rate or other features.

The approach with exposing cells to hypoxic conditions should also be better justified in the introduction

The most important flaw that has to be corrected is lack of statistical analysis.

Results of pluripotency markers expression are interesting but description of the comparison between WJMSCs and iPS should be more explicit and emphasize for example thousand fold lower expression in WJMSCs. What does it mean “expression of both factor in iPS cells exceeded 90%”? 90% of cells expressing these factors or expression level was at 90% of some reference value? Please express that more clearly. Is it that increased level of NANOG is primarily a result of more cells expressing nanog at these very low levels or expression level in some cells goes up to levels seen in pluripotent cells? Immunocytochemistry in Fig.3 actually do not seem to support flow cytometry results with immunocytochemistry showing only few cells highly expressing nanog. Please explain this discrepancy.  

There are no references in text of the results to subpanels D and E of the Figure 3. Are these duplicates of data shown in panels B,C?

Presentation of bioluminescence data is somewhat confusing. First of all photon flux (photons/s) should be quantified with mean+/-SEM over time reported to draw any conclusions. Measurement timepoints are different for each group of just different timepoints were selected for this figure? To better utilize space of this figure and clearly present the data representative animals should be displayed for each group and for each key measurement time. Consistency and group to group comparison will be shown on quantification graph.

The fact that cells were ultimately lost in these immune deficient mice is concerning and should be discussed.  Survival of cells for 37 days is short; would that disqualify these cells from therapeutic applications?

 Statement that pluripotency markers are expressed by MSCs at relatively high level could be misleading. Expression level is few orders of magnitude lower compared to pluripotent cells. Please rephrase.

A paragraph about potential applications should be included in the introduction the discussion section.

Authors on several occasions refer to previous results without citing relevant papers

255- 267 there is no information how many umbilical cord samples were used 

368-383 Is repetition of 4.5 subparagraph

Figure 1D charts are to small and unreadable, also part of the caption could be moved to results section

Figure 2C again is to small and unreadable

Author Response

The article addresses an interesting issue of the importance of pluripontency markers for the function of Wharton’s Jelly-derived mesenchymal stem cells that in fact by definition are not pluripotent. Due to the advantages of this cell source (ease of access, high potency) there is ample interest in utilizing them clinically. Overall this paper is informative and conclusions are in most cases supported by presented data several deficiencies should be corrected to improve clarity and scientific value of this work. Specific comments are listed below:

1.At the end of introduction authors indicate WJMSCs as more potent than adult stem cells. Please specify whether it is differentiation repertoire, proliferative rate or other features.

This part of the publication has been changed according to the recommendation of the reviewer. We have added statement describing unique properties of WJ-MSCs as cells derived from fetal tissue.

2.The approach with exposing cells to hypoxic conditions should also be better justified in the introduction

According to the recommendation of the reviewer we have elucidated the reason of exposing cells to hypoxic conditions.

3.The most important flaw that has to be corrected is lack of statistical analysis.

The statistical analysis has been performed.

4. Results of pluripotency markers expression are interesting but description of the comparison between WJMSCs and iPS should be more explicit and emphasize for example thousand fold lower expression in WJMSCs. What does it mean “expression of both factor in iPS cells exceeded 90%”? 90% of cells expressing these factors or expression level was at 90% of some reference value? Please express that more clearly. Is it that increased level of NANOG is primarily a result of more cells expressing nanog at these very low levels or expression level in some cells goes up to levels seen in pluripotent cells? Immunocytochemistry in Fig.3 actually do not seem to support flow cytometry results with immunocytochemistry showing only few cells highly expressing nanog. Please explain this discrepancy.  

The expression level all of markers was estimated with reference to appropriate isotype controls. 90% means that 90% of cells express given marker, for each analysis we took 10 000 events per gate.

The expression level of NANOG is about 20% estimated by flow cytometry. Fig.3 – our microscopic observations confirm the results obtained from cytometry, we have observed numerous cells expressing NANOG, some apparent discrepancies arise probably from camera capabilities.

5.There are no references in text of the results to subpanels D and E of the Figure 3. Are these duplicates of data shown in panels B,C?

Fig.3 D, E show the immuno-staining for NANOG and OCT-4 in modified cells (WJ-MSCs Luc +) to show no differences in the expression of the these markers in cells after modification.

6.Presentation of bioluminescence data is somewhat confusing. First of all photon flux (photons/s) should be quantified with mean+/-SEM over time reported to draw any conclusions. Measurement timepoints are different for each group of just different timepoints were selected for this figure? To better utilize space of this figure and clearly present the data representative animals should be displayed for each group and for each key measurement time. Consistency and group to group comparison will be shown on quantification graph.

The main aim of fig. 4 is to show how the cells behave after injections. The most important information  resulting from this fig. is that WJ-MSCs don’t form a tumors. Our intention wasn’t  to show what is viability the cells after administration. Measurement timepoints were the same in each group, but for this figure we have choose different timepoints to show how the signal changed during the experiment.

7.The fact that cells were ultimately lost in these immune deficient mice is concerning and should be discussed.  Survival of cells for 37 days is short; would that disqualify these cells from therapeutic applications?

This part of the publication has been discussed according to the recommendation of the reviewer.

8. Statement that pluripotency markers are expressed by MSCs at relatively high level could be misleading. Expression level is few orders of magnitude lower compared to pluripotent cells. Please rephrase.

This part of the publication has been changed according to the recommendation of the reviewer.

9.A paragraph about potential applications should be included in the introduction the discussion section.

A paragraph about potential applications has been included in the introduction of the discussion section.

10.Authors on several occasions refer to previous results without citing relevant papers

11.255- 267 there is no information how many umbilical cord samples were used 

The information about the quantity of umbilical cords samples has been added according to the recommendation of the reviewer.

12.368-383 Is repetition of 4.5 subparagraph

This repetition has been removed from the publication.

13.Figure 1D charts are to small and unreadable, also part of the caption could be moved to results section

The histograms have been changed to more readable. The part of the caption has been modified and moved to results section.

14.Figure 2C again is to small and unreadable

 The histograms have been changed to more readable.